# Practice of Noseband Use and Intentions towards Behavioural Change in Dutch Equestrians

**DOI:** 10.3390/ani9121131

**Published:** 2019-12-12

**Authors:** E. Kathalijne Visser, Monique M. F. Kuypers, Jennifer S. M. Stam, Bernd Riedstra

**Affiliations:** 1Department of Applied Research, Aeres University of Applied Sciences, P.O. Box 374, 9250 AJ Dronten, The Netherlands; 2Behavioural Biology, Groningen Institute for Evolutionary Life Sciences, University of Groningen, Nijenborgh 7, 9747 AG Groningen, The Netherlands

**Keywords:** horse welfare, equestrian sport, noseband, human behaviour change, attitudes

## Abstract

**Simple Summary:**

The space between the noseband and the skin of competition horses is a current welfare issue. The practices regarding the noseband tightness of Dutch horses was studied, as well as the intentions of Dutch equestrians when fastening the noseband. More than half (59%) of Dutch riders had their horses’ nosebands tightened according to the new two-finger regulation, implemented 1 April 2019. Dressage horses and older horses wore less tight nosebands compared to show jumping and younger horses. Results of an internet survey on intentions for noseband use showed that 54.5% of the respondents agreed with the new regulation and 62% believe that it will improve horse welfare. The respondents could be categorised into three different groups that differed regarding their own attitude towards noseband tightening behaviour, how peer pressure may affect noseband tightening behaviour, and how they perceived the new regulation. To improve horse welfare, knowledge transfer should include different strategies for different groups. Moreover, to convince equestrians to adhere to, and ensure a successful implementation of, the new regulation, transparency and objective measurements should be put in place.

**Abstract:**

Understanding equestrians’ noseband tightening practices and intentions is necessary to target welfare improvement strategies. Firstly, we measured tightness in dressage and show jumping horses in The Netherlands, shortly after implementation of the two-finger rule by the Royal Dutch Equestrian Federation. Noseband tightness decreased with age, was less tight in dressage horses than in show jumpers, and was dependent on the interaction between competition level and discipline. Fifty-nine percent of the riders tightened nosebands to such an extent that they adhered to the new regulation. Secondly, we conducted an online survey to gain insight into whether riders were aware of noseband use and tightening behaviour. Of the 386 respondents, 54.5% agreed with the new regulations, and 62% believe that it improves horses’ welfare. Applying cluster analysis to statements regarding their own attitude, peer pressure, and behavioural control produced three clusters. Noticeably, a lower percentage of Cluster 1 respondents (38%) performing at higher levels was convinced that the new regulation improved welfare than Cluster 2 (77.9%) and 3 (89.0%) respondents. Designing strategies to ensure the successful implementation of the new regulation and to convince equestrians to comply would be most effective if targeted differentially, and should include a transparent and objective form of regulation.

## 1. Introduction

In recent decades, the use of animals for consumption, sport, and leisure has seen an increase in regulations aimed at improving animal welfare. The horse is a species used for all these purposes and, in order to obtain a social license to use them for sports or leisure, the equestrian industry has to prove that horse welfare is protected. Since 2012, the use of the noseband for horses, with respect to how tightly it is applied, has received considerable attention [1,2,3,4]. 

The noseband is part of the bridle, which is the main instrument used to control horses by riders. It is attached to the head, one of the most sensitive parts of a horse’s body. By applying pressure on the horse via the bridle, a rider tries to control a horse through both their training and the restraint brought about by this bridle [2]. The nosebands on the bridles can be applied with varying levels of tightness. A reason for applying a particular level of tightness of the noseband in, e.g., dressage may include preventing or discouraging a horse from opening its mouth. In doing so, one may improve the observers’ (in this case, the judges’) impression of the level of “submission” of the horse. Submission is desirable in dressage because marks are deducted if the horse is judged to be resisting the bit [5]. Another reason for noseband tightening is increased control over the horse. This is supported by evidence that, with tighter nosebands, riders need to use less rein tension, however transient, to decelerate their horses [6]. However, when there is no space between the noseband and the nasal plane, horses show signs of physiological stress: a significantly increased heart rate and decreased heart rate variability [3]. Also, the post-inhibitory behavioural rebound response (significantly increased yawning, licking and swallowing) during a recovery phase (after removing the noseband) were indicative of compromised welfare [3]. Additionally, post-competition measurements in Denmark showed a correlation between oral lesions and noseband tightness for certain noseband types [7]. Increased tightening of the upper noseband significantly increased the incidence of oral lesions (10.79% (noseband tightness < 2 cm) vs. 7.51% (noseband tightness between 2 and 3 cm)). Loosening the upper noseband decreased the prevalence of oral lesions by 34% (odds ratio 0.66). Choosing a certain noseband type and applying the right level of tightness therefore has implications for animal welfare. 

There are a wide variety of nosebands commercially available. The most commonly used noseband at elite levels of equestrian competition in dressage, where a double bridle is mandatory, is the Swedish cavesson, more commonly called the crank noseband [2]. This noseband provides the user with a mechanical advantage to tighten the noseband more easily by way of a leveraged pulley system [2,8]. The crank noseband allows for a doubling of the tightness for a given amount of handler tightening effort [2], which may have welfare implications for the horse [7]. The use of these crank nosebands and extreme tightening of the noseband is thought to be increasing in equestrian sport [4], and is therefore likely to have an impact on the welfare of horses.

The International Equestrian Sport Federation (FEI), and countries such as Germany, Denmark and New Zealand recently introduced new competition regulations on how tightly nosebands may be applied during competition. A recent study found that, in Belgium, Ireland, and the UK, 44% of the 750 horses studied (eventers, dressage, and performance hunter horses) wore nosebands that allowed no space for a measuring gauge to be inserted between the noseband and the nasal plane, 7% had space to accommodate half a finger, 23% had space for one finger, and 19% for 1.5 fingers [2]. Only 7% of nosebands allowed for the placement of the equivalent of at least two fingers (which is about 1.5 cm) [2]. Incidence of lack of space between the noseband and the skin over the nasal plane was largest in eventers, followed by dressage horses. In contrast, in Germany, more nosebands (69.86%) were tightened according to the two finger rule [9]. Here, dressage riders had their nosebands significantly (36.5% of the riders) tighter than the jumpers (18.42% of the riders). In a study of Doherty, et al. [2], the authors concluded that the practice of measuring the space between noseband and nasal plane does give a good indication of current noseband use, hence they highlight the need for further research into riders’ motivations for tightening nosebands excessively. 

In The Netherlands, as of 1 April 2019, the Royal Dutch Equestrian Federation (KNHS) has implemented a new competition regulation, which states that there must be at least a space of 1.5 cm between the noseband and frontal nasal plane. Implementing new regulations often requires a change in behaviour by those affected. In human psychology, the Theory of Planned Behaviour (TPB) is often used to explain or to predict how humans will or will not change their future behaviour. In short, the theory links one’s beliefs with behaviour. The theory states that attitudes toward behaviour, subjective norms, and perceived behavioural control together shape an individual’s behavioural intentions and behaviours. The attitude toward behaviour refers to the degree to which a person has a favourable or unfavourable evaluation or appraisal of the behaviour in question. The subjective norm, the social factor, refers to the perceived social (peer) pressure to perform or not to perform this behaviour. The perceived behavioural control is the ease or difficulty of performing the behaviour and is assumed to reflect past experience as well as anticipated impediments and obstacles, which also influence intention [10]. In an explorative study on diversity in horse enthusiasts regarding horse welfare issues, it appeared that horse enthusiast can be categorised into four different groups regarding their believes and motivations for improving horse welfare in practice [11]. 

Currently, there is a lack of information on noseband tightness levels in The Netherlands and the attitudes of equestrians towards these new regulations. In this study, we therefore firstly present data on noseband tightness, measured during several national competitions in The Netherlands (2019), shortly after the implementation of these new regulations. Furthermore, we analysed noseband tightness levels in relation to discipline, competition level, and age of horse and rider. At the same time, we issued an internet survey aimed at evaluating the motives towards the use of nosebands. Following this, we performed a cluster analysis of the respondents’ intentions to change noseband tightening behaviour and analysed this in relation to their discipline and practice.

## 2. Material and Methods

### 2.1. Measurements Level of Tightness of the Noseband

The tightness of nosebands was measured in 100 horses competing at four different national competitions in The Netherlands during May 2019. Half of the horses were competing in dressage, the other half in a show jumping competition. The age of the horses was 9 (± 3.4) years. The competition levels were B (Preliminary), L (Novice), M (Elementary) and Z (Medium). Only horses for which the owner/rider gave an informed consent were included in the study. Besides tightness of the noseband, we recorded the following variables: type of noseband used, competition level, discipline, age of the rider, and age of the horse.

Traditionally, controlling the tightness of nosebands by officials or riders is performed by placing fingers between the noseband and the frontal nasal plane [12]. In 2016, the International Society for Equitation Science (ISES) produced a taper gauge to systematically measure the tightness of the noseband. In the current study, the level of tightness was measured by one of the authors (JSMS) using this ISES taper gauge. Counterintuitively, the higher the level of tightness, the more loosely a noseband is applied. For a correct measurement, the taper gauge needs to be under the noseband, in a rostro-cranial direction, as far as it would progress without causing dorsal displacement of the noseband on the nose, or elevation of the horse’s head. It then will indicate the space between the noseband and frontal nasal plane in a category resembling the number of fingers that can be put between the noseband and the frontal nasal plane: 0, 0.5, 1, 1.5, 2 or more than 2 fingers. A measurement of two fingers equals a 1.5 cm space. 

### 2.2. Online Survey

To gain insight into the extent to which riders are aware of the use of the noseband and their tightening behaviour, an online survey was conducted among Dutch equestrians. The following topics were investigated: How does an equestrian determine the tightness of the noseband? What does the equestrian think are the pros and cons of a tight noseband? What is the knowledge of the equestrian of the (new) rules and regulations with respect to the tightness of the noseband, and to what degree does peer pressure play a role in the tightness of the noseband? The survey contained 13 questions (see Table 4); three questions were set-up with a total of 18 propositions on a 1 (totally disagree) to 5 (totally agree) scale and were focussed on the attitude, social factors, and perceived behavioural control regarding noseband use and tightness of nosebands. The survey was distributed by social media, with a bias towards dressage Facebook pages. 

### 2.3. Statistical Analysis

#### 2.3.1. Noseband Tightness Levels of Horses during Competition Events

In order to avoid over-parametrisation and exclude variables in our Generalized Linear Model (GLM) approach (see below), we first explored the relatedness of the variables noseband tightness level, age of the horse, age of the rider, and competition level in both disciplines (dressage and show jumping) separately using Spearman Rank correlations. This non-parametric approach was used because none of the variables were distributed normally (Shapiro–Wilk: all W-values > 0.73, all *p*-values < 0.0001). We performed a GLM (ordinal probit) with noseband tightness as a dependent variable, discipline as an independent factor, and competition level and age of the horse as covariates. Our initial model included the main factors and all two- and three-way interactions. We then proceeded, excluding interactions and main factors that did not contribute to explaining variation in noseband tightness from the model. Finally, for comparative purposes, we performed post hoc analyses on the age of the horse, age of the rider, and competition level, using Wilcoxon rank sum (WRS) tests. Alpha was set to 0.05 in all test.

#### 2.3.2. Internet Survey: Attitudes toward Noseband Usage

A cluster analysis was performed to identify relatively homogeneous segments of equestrians regarding their motives for noseband use. The cluster analysis included only those respondents (N = 386) that had answered all 18 propositions in the survey (see Table 4, Results Section) relating to their motives for practices regarding noseband use. A two-step clustering procedure was performed. The first step was the hierarchical clustering method, Ward’s method with Euclidean distances, and aimed to define the number of clusters. The second step, the K-means clustering, is a non-hierarchical procedure, which further reduced the heterogeneity within the clusters in order to get a more accurate cluster membership. A 3-cluster solution was determined to be the most appropriate to identify these 386 respondents. 

#### 2.3.3. Internet Survey: The Link between Intentions and Practice 

For further analysis, only respondents that had indicated that they performed in dressage, show jumping, eventing, or a combination of those were included; hence, three discipline categories were constructed: (1) People that exclusively performed in dressage, (2) people exclusively performing in show jumping, and (3) people that performed in both. People that indicated that they performed in eventing were included in the latter group. 

Firstly, we analysed the respondents’ opinion, whether they agreed with the new regulation on noseband tightness and whether they thought this regulation would improve the horses’ welfare using a GLM with agreement regulation (yes/no) and welfare improvement (yes/no) as dependent variables, discipline (three levels), competition level (five levels) and cluster (three levels) as independent factors, and age of the respondent as a covariate. To avoid over-parametrisation, we only analysed the main factors, excluded all two- and three-way interactions and proceeded with the analyses as described above. We then performed post hoc analyses (Wilcoxon rank sum or Kruskal–Wallis one-way ANOVA) on those main factors in the model that contributed to explaining variation in a dependent variable.

Secondly, in order to compare the survey responses with the results of the actual measurements of noseband tightness and type of noseband used in 100 horses, we used GLMs with tightness level (three levels: 0, 1, 2 fingers) as indicated in the survey and noseband type (two levels, as described above) as independent variables, cluster (three levels), competition level (five levels), discipline (three levels), agreement with the regulation (two levels), and thinking it improves welfare (two levels), and age of the respondent as a covariate. Here, we also only analysed the main factors, and excluded all two- and three-way interactions and proceeded with the analyses as described above. We again performed post hoc analyses (Wilcoxon rank sum or Kruskal-Wallis one-way ANOVA) on those main factors in the model that contributed to explaining variation in a dependent variable. Finally, we re-executed the GLM procedure with noseband tightness as an independent variable, but now excluded respondents competing at levels exceeding Z, pooled the disciplines show jumping and eventing, and added the interaction between discipline and competition level. All statistical analyses were performed with SPSS25 and Statistix 8.0.

## 3. Results

### 3.1. Noseband Tightness of Horses during Competition Events.

In 37 dressage horses and 22 show jumping horses, the noseband tightness level was two fingers and, in total, 71 horses had a noseband tightness level larger than or equal to 1.5 fingers (see Figure 1). In two out of the 100 cases, we observed a noseband tightness level of zero fingers; both cases concerned show jumping horses. In both disciplines, older horses had higher noseband tightness levels (scores), i.e, more loosely applied (space under the) nosebands, than younger horses (Spearman Rank: dressage; ρ = 0.31, *p* = 0.0293, show jumping ρ = 0.31, *p* = 0.0278). Interestingly, the age of show jumpers, not dressage horses, increased with increasing competition level (Spearman Rank show jumping: ρ = 0.42, *p* = 0.0025; dressage: ρ = 0.11, *p* = 0.4538), and, in show jumping, noseband tightness level decreased with increasing competition level (Spearman Rank: ρ = −0.30, *p* = 0.0357); again, this did not occur in dressage horses (ρ = 0.18, *p* = 0.2020). There was no relationship between noseband tightness level, age of the horse, or competition level with the age of the rider (Spearman Rank: all ρ-values ranged between −0.12 and 0.2, all *p*-values > 0.16).

Overall, there was a significant interaction between competition level and discipline on noseband tightness level (GLM: Wald χ^2^ = 6.1, *p* = 0.013); the level of tightness decreased more with increasing competition level in show jumping horses than in dressage horses (see Figure 2). 

Overall, there was a near significant decrease with increasing competition level (GLM: Wald χ^2^ = 3.2, *p* = 0.072), levels of tightness increased with age (GLM: Wald χ^2^ = 11.6, *p* = 0.001), and were higher in dressage horses than show jumping horses (GLM: Wald χ^2^ = 4.6, *p* = 0.033). Post hoc analyses showed that the observed dressage horses were older than the show jumping horses, but performed at a lower competition level (see Table 1). There was no difference between the age of their riders. 

### 3.2. Types of Nosebands Used during Competition Events

In total, we identified eight types of nosebands that were in use (see Table 2). Two of these types (cavesson crank nosebands) have a leveraged closure mechanism, allowing for a tighter, more precise fit. We therefore pooled the types used based on this functionality and tested whether the frequency of use differed between the disciplines using a proportion test.

There was a marked difference between the disciplines in the type of noseband used: overall, 41 out of 50 (82%) dressage horses were fitted with cavesson crank nosebands, whereas only three of the 50 (6%) show jumping horses were fitted with this type (proportion test: Z = 7.45, *p* < 0.0001). 

### 3.3. Internet Survey: Motives for Noseband Usage

The age of the respondents varied, ranging between 11 and 67 years of age (30.0 ± 12.2); most respondents were active in dressage (77.3%), followed by show jumping (14.0%). Dressage riders used a cavesson crank noseband with flashband (42.7%), followed by the cavesson noseband with flashband (22.4%). Most show jumpers used a cavesson noseband with (36.5%) or without (19.2%) flashband. The cluster analysis revealed three clusters. Cluster 1 accounted for 42.0% of the respondents, Cluster 2 for 31.3% and Cluster 3 for 26.7%. Cluster 1 included the highest percentage of dressage riders competing at (advanced) medium levels (level Z or ZZ, see Table 3).

Applying the Theory of Planned Behaviour (TPB) to the results of the cluster analysis shows that, concerning the ‘attitude’ (statements 1–8), the respondents in Cluster 1 believe that riding with a tightly fitted noseband is better for the welfare of a horse, whereas respondents in Clusters 2 and 3 disagree with this. With regard to the ‘perceived behaviour control’, results show that Cluster 1 and Cluster 3 respondents agree that the tightness of the noseband should be measured with a measuring device instead of using two fingers. However, Cluster 1 respondents argue that the standard for noseband tightness should be adjusted to one finger, while Cluster 3 respondents argue that the standard should remain two fingers. With regard to the ‘social factors (peer pressure)’, there are marked differences between the clusters. Only respondents in Cluster 1 would not approach someone that tightened the noseband of the horse too tight, whereas only respondents in Clusters 2 and 3 would care if someone told them that their noseband was fastened too tight. Moreover, only respondents in Cluster 3 would not tighten the noseband faster if an instructor told them to do so (see Table 4). 

### 3.4. Internet Survey: Intentions and Practice 

A large majority (295/301 = 98%) of respondents were aware of the current new regulations concerning noseband tightness, of which 136 (46%) had actually looked up the regulations in the statutes of the KHNS. Furthermore, 164 (54.5%) agreed with the new regulations, and 186 (62%) believe that it improves the horses’ welfare. Agreement with the rules depended on cluster and competition level (GLM: χ^2^_cluster_ = 74.6, *p* < 0.001, χ^2^_comp. level_ = 21.0, *p* < 0.001), not on discipline or age of the respondent (GLM: χ^2^-values < 1.1, *p*-values > 0.58). Fewer Cluster 1 people (34/142 = 27.4%) were in agreement with the regulations than Clusters 2 (66/86 = 76.7%) and 3 (64/73 = 87.7%) (Kruskal–Wallis one-way ANOVA: H = 102.7, *p* < 0.0001). Respondents competing at levels B, L and Z were in higher agreement (92/144 = 63.8%) with the regulations than respondents performing at higher levels (33/104=31.7%), whereas people performing at M levels scored intermediate (30/53 = 56.6%). Concerning opinion towards welfare improvement by the new noseband tightness regulations, this solely depended on cluster (GLM: χ^2^_cluster_ = 57.4, *p* < 0.001), not on age of the respondent, competition level, or discipline (GLM: χ^2^-values < 5.1, *p*-values > 0.27). Cluster 1 respondents (54/142 = 38%) were much less convinced that the new regulations improved welfare than Cluster 2 (67/86 = 77.9%) and 3 (65/73 = 89.0%) respondents (Kruskal–Wallis one-way ANOVA: H = 66.2, *p* < 0.0001).

### 3.5. Type of Noseband 

A minority (4%) of respondents indicated that their noseband of choice was recommended by a store, by a trainer, or because a well-known rider used it. Thirty one percent indicated that it was not a specific choice, 45.7% indicated that it was simply the most pleasant one to use, and, finally, 19.3% had different (unknown) reasons. Over half of the respondents (161/300 = 54%) used the cavesson crank noseband. The use depended on discipline (GLM: χ^2^ = 26.5, *p* < 0.001), not on competition level, whether or not respondents agreed with the regulation, whether or not they believed the regulation would improve welfare, nor on cluster (GLM: All χ^2^- values <4.9, all *p*-values > 0.3). In agreement with the actual measurements we took during the competitions (see above), this type was most commonly used in dressage (60.6%) and it was used infrequently in show jumping (20%) or when respondents performed in both disciplines (17.2%, Kruskal–Wallis one-way ANOVA: H = 29.3, *p* < 0.0001). 

### 3.6. Noseband Tightness Levels

Also in line with the actual measurements we took, only a small number of respondents (19; 6.3%) indicated that they applied a tightness level of 0 fingers, 134 respondents (44.7%) a level of one finger, and 147 respondents (49%) a level of two fingers. In contrast to the actual measurements, tightness did not depend on discipline or competition level (GLM: χ^2^-values < 3.2, *p*-values > 0.3), but did on cluster, whether or not respondents agreed with the regulations, and whether or not they thought the regulations would improve welfare (GLM: χ^2^_cluster_ = 20.9, *p* < 0.001, χ^2^_agreeing_ = 26.0, *p* < 0.001, χ^2^_welfare_ = 12.7, *p* < 0.001). Fifty two percent of the respondents indicated that they actually checked the amount of space between the noseband and the nasal plane, all of them using their fingers, none of them using a special device. The remaining respondents indicated that they applied the noseband by intuition (34.3%) or by some other method (2.6%). 

Cluster 1 respondents applied a lower tightness level (median + 1st and 3rd quartile in brackets: 1 (1–1) compared to people in Cluster 2 and 3 (both 2 (1–2); Kruskal–Wallis one-way ANOVA: H = 98.6, *p* < 0.0001). Respondents agreeing with the new regulations on noseband use applied a higher level of tightness (i.e., more space between the nasal plane and noseband) than respondents not agreeing with the new regulation (median + 1st and 3rd quartile in brackets: 2 (2–2) vs. 1(1–1), Wilcoxon rank sum: Z = 11.1, *p* < 0.0001). Respondents that believe the regulations enhance welfare applied a higher level of tightness (more space) than respondents that do not believe so (median + 1st and 3rd quartile in brackets: 2 (1–2) vs. 1 (1–1), Wilcoxon rank sum: Z = 9.7, *p* < 0.0001). 

## 4. Discussion

Our data on the noseband tightness levels of horses in dressage and show jumping competitions at the national level in The Netherlands shows that the majority (59%) had their nosebands tightened to such an extent that is was possible to insert the ISES taper gauge under the noseband to at least the two fingers tightness, as the regulations prescribe. The authors are not aware if, at either one of these competitions, the noseband tightness was checked by an official. Although in Belgium, Ireland, and the UK the amount was only 7% [2], Dutch tightness levels closely resembled German levels, where about 70% of horses at small, national leisure competitions had a noseband fit of two or more fingers [9]. However, compared to that study, show jumping horses had significantly tighter nosebands than dressage horses, whereas in Germany this was reversed. To date, no information on genetic line and temperament has been collected in the studies on noseband use. This may be an important omission, since breeding for a certain discipline implies a preference for certain breeds, genetic lines, and, hence, temperament. This may vary between countries and may systematically invoke riders to apply a certain level of tightness and associated choice of noseband type.

As mentioned before, certain types of nosebands facilitate tightening of the noseband more easily than others. Surprisingly, even though the nosebands with this capacity were much more commonly used in dressage horses than in show jumping horses, noseband tightness levels were in fact higher (more space) in Dutch dressage horses than show jumping horses. Both the dressage horses we measured, as well as those in the internet survey, were fitted mainly with the cavesson crank noseband. In the study of Doherty, et al. [2], the type of noseband was also included in their analysis, however, they did not differentiate between different types of cavesson nosebands, meaning that the crank nosebands were included in the cavesson category. We pooled the crank nosebands in one category and all the others, including the flash nosebands and drop noseband, in another category. 

The control of a horse is inherently more challenging when a horse is ridden at speed towards and over fences, as compared to a dressage performance. Therefore, the choice of noseband and noseband tightness may differ between disciplines. Moreover, nosebands are sometimes worn solely for aesthetic purposes [7] and, since dressage horses are judged on total appearance, dressage riders are likely to choose a popular type of noseband. The reason respondents gave in our survey for the choice of noseband was not clear cut. In our study, there was a strong bias in the use of these nosebands between different disciplines, which may be culturally determined and may also vary between countries. Further research is needed to understand the different motives for noseband use between disciplines.

In any case, the crank noseband has a leveraged closure mechanism, allowing for a tighter, more precise fit. To function effectively as a riding aid, the noseband needs to provide a bit of space between the nasal plane and noseband. When there is this bit of space and the horse opens its mouth, the noseband tightens and puts pressure on the horse’s nasal plane. When the mouth closes, the pressure is relieved immediately [7]. However, the use of bit, spur and noseband may cause lesions, undoubtedly causing discomfort for the horse [7]. Comparing different types of nosebands, including the cavesson, Uldahl and Clayton [7], showed that lesions at the commissures of the lips were related to the tightness of the upper strap of the noseband. There was a significantly higher risk of lesions when the upper noseband was tightened allowing less than two fingers’ space, compared to two to three fingers. Because, in our study, we did not differentiate between categories of more than two fingers’ space and we did not inspect for lesions, we cannot conclude that even more space than two fingers would benefit horse welfare.

We also found that the space between noseband and nasal plane depended on the horse’s age and the competition level the horses were performing at. While the space remained more or less constant with increasing competition level in dressage horses, it decreased in show jumping horses: Show jumping horses at higher competition levels had tighter nosebands compared to show jumping horses competing at lower levels, which may be caused by subjective feeling or need for increased control over the horse when fences become higher. However, in the German study [9], no such effect was found. Moreover, in contrast to our study, Doherty, et al. [2] did not find an effect of age on tightness among eventers, dressage horses and performance hunters, whereas we found that older horses had significantly more space between the noseband and nasal plane, which may be a consequence of a longer training and an increased level of trust in the relationship between rider and horse.

There seems to be a contradiction in our results, as one may expect that it requires longer preparation to perform at higher levels, and that horses therefore competing at higher levels would be older. This was the case for show jumpers but not dressage horses. Our sample therefore may have been heterogeneous in the sense that we accidentally sampled more talented young show jumping horses performing at higher levels, whereas the dressage subsample contained less talented (and therefore competing at lower levels) but older horses. It would therefore be informative to include the competition rankings in future analyses.

Our internet survey showed that, even though the regulation had been implemented only one month before, 98% of the respondents were aware of the new regulation. The respondents, however, were not chosen randomly, the survey distribution was mainly aimed at dressage riders, and moreover there may be an overrepresentation of respondents interested in noseband use which may have caused this high percentage. As with the measurements at the events, most dressage respondents used the cavesson crank noseband. Over half of the respondents (52%) checked the width between the noseband and nasal plane with their fingers and 34% fastened the noseband on intuition without checking. 

Respondents could be assigned to one of three clusters, based on their attitude towards noseband use, peer pressure, and their perceived behavioural control. The first clusters’ intentions towards noseband use were particularly different from those of Cluster 2 and 3 respondents. Only 27.4% of Cluster 1 respondents agreed with the new regulation and they were much less convinced that it would benefit horse welfare than the other respondents. Furthermore, Cluster 1 respondents were competing at higher levels compared to Cluster 2 and 3 respondents. Not surprisingly, and in line with the actual measurements, respondents in Cluster 1 indicated that they fastened the noseband tighter compared to respondents in Cluster 2 or 3. 

For a behavioural change, several preconditions need to be in place, namely the attitude, the social factor, and perceived behavioural control [10]. In this current study, all of these factors seem to play a crucial role when comparing Clusters 1 respondents to Cluster 2 and 3 respondents. For instance, if Cluster 1 respondents do not feel that this new regulation benefits horse welfare or that it is better for overall performance, then the intention to change their tightening behaviour is reduced. Similarly, if respondents in Cluster 1 feel that they only have a limited behavioural control, because an objective measurement to check for space between the noseband and nasal plane is not implemented, they will have less intention of changing their behaviour. Numerous studies have examined the opinions and perceptions of horse enthusiasts related to horse welfare [11,13,14]. It is a common phenomenon that people are more concerned about the welfare of horses of other groups (professionals versus non-professionals [11], or disciplines [14]) than the welfare of their own horses. Therefore, in the current study we not only included respondents’ own attitude, but also the effect of a social factor, namely peer pressure. Results showed that especially Cluster 1 respondents were less likely to be affected by the opinion of other riders about noseband tightening. In addition, these respondents would not approach other riders if they fastened nosebands too tight.

Given that the new regulation improves horse welfare, because of the increased risk of oral lesions [7], a profound physiological stress response, and deprivation of oral behaviours [3] when nosebands are fitted more tightly, communicating this knowledge more clearly, especially to Cluster 1 respondents, seems likely to be most effective. However, (Cluster 1) respondents competing at higher levels may be reluctant to change their behaviour, even when they are convinced it improves welfare, if this change in behaviour results in a loss in competitive position. This could be easily resolved by instructing, e.g., dressage judges, not to deduct points for horses ‘resisting the bit’ and by a transparent and objective practice of checking noseband tightness directly before and after competition performances. 

## 5. Conclusions

The results of our study confirm the findings in Germany in that, as required by the new regulation, more than half of the Dutch horses had their nosebands tightened, leaving at least two fingers space between the noseband and nasal plane, and that the discipline, age, and performance level of the horse affect the noseband tightness. The intention and motivation for riders to change their tightening behaviour differs and is related to their attitude towards the new regulation, welfare benefits, peer pressure, and perceived behavioural control. More insight is needed into the intentions of equestrians of different disciplines regarding tightening behaviour in order to convince riders to adhere to, and ensure a successful implementation of, the new regulation that improves horse welfare. 

## Figures and Tables

**Figure 1 animals-09-01131-f001:**
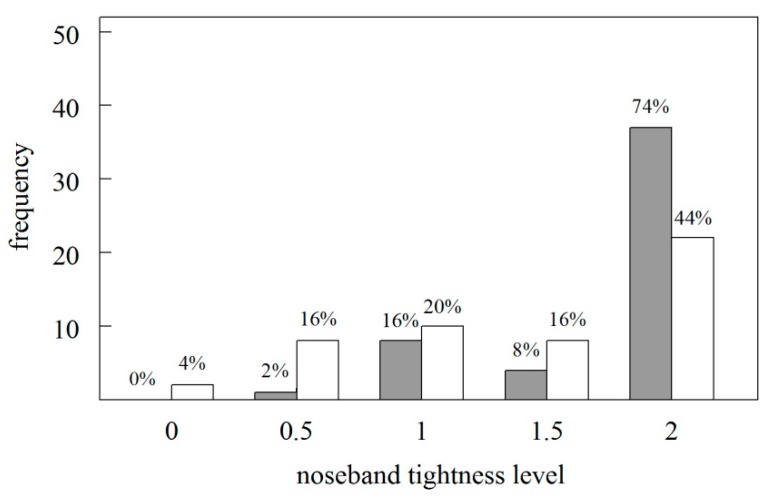
Distribution and percentage of noseband tightness levels in dressage (filled bars) and show jumping horses (open bars) used in this study. The tightness of the noseband is inversely related to the unit of measurement recording the level of tightness (fingers), meaning that there is an increasing amount of space between the noseband and the skin from 0 to 2 (see Section Material and Methods). Noseband tightness level was significantly higher in dressage horses.

**Figure 2 animals-09-01131-f002:**
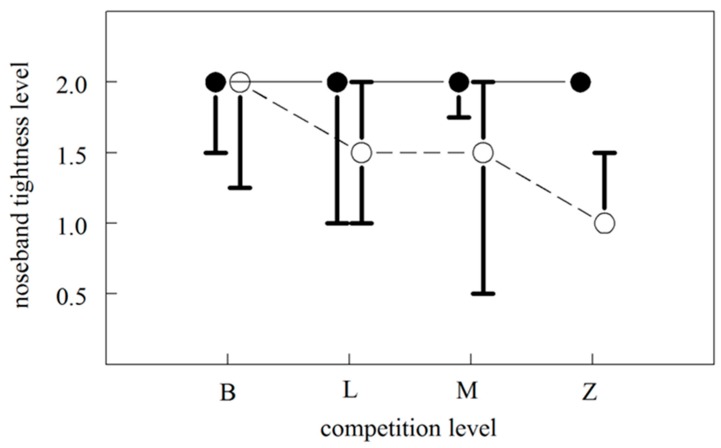
The relationship between competition level (increasing in performance level from B (Preliminary) to Z (Medium)) and noseband tightness in dressage horses (filled dots) and show jumping horses (open dots). Dots indicate the median tightness level which ranges from 0 (very tight) to 2 (loose). The bars indicate the 1st and 3rd quartile. The level of tightness depended on the interaction between competition level and discipline.

**Table 1 animals-09-01131-t001:** Median (1st and 3rd quartile) noseband tightness level (where an increasing number indicates more space between the noseband and the nasal plane) of 50 dressage horses and 50 show jumping horses in The Netherlands, measured with the ISES Taper Gauge; their median (1st and 3rd quartile) competition level [going from B (Preliminary), to L (Novice), M (Elementary) and Z (Medium) in increasing performance level]; and age of the horses and riders. Z- and *p*-values refer to the Wilcoxon rank sum (WRS) tests performed on the variables in the first column.

Group Characteristics	Dressage	Show Jumping	Z	*p*
Noseband Tightness Level	2 (1.5–2)	1.5 (1–2)		
Competition Level	M (L–Z)	L (B-M)	2.479	<0.001
Age of Horse	10 (8–13)	7 (5.8–9)	4.273	0.013
Age of Rider	27 (20.8–39.5)	26 (20.8–33.0)	0.369	0.712

**Table 2 animals-09-01131-t002:** Types of nosebands used during national Dutch competition events in May 2019 on dressage and show jumping horses separately (for figures of different noseband styles, see [7]).

Type of Noseband	Dressage (N = 50)	Show Jumping (N = 50)
Cavesson noseband without flashband	1	0
Cavesson noseband with flashband	2	36
Cavesson crank noseband without flashband	7	1
Cavesson crank noseband with flashband	34	2
Drop noseband	3	1
Micklem	2	2
Mexican noseband	0	5
Other	1	3

**Table 3 animals-09-01131-t003:** Distribution (in percentages) of respondents over the clusters regarding age, discipline and dressage competition level (B = Preliminary, L = Novice, M = Elementary, Z = Medium, ZZ = Advanced Medium).

Group Characteristics	Cluster 1	Cluster 2	Cluster 3
Age (years)	29.8	30.5	30.2
Discipline: dressage (%)	39.9	24.4	20.4
Discipline: show jumping (%)	6.0	5.5	3.7
Dressage competition level B-M	14.2	13.2	12.5
Dressage competition level Z-ZZ:	32.9	15.6	11.5

**Table 4 animals-09-01131-t004:** Characterization of clusters based on the respondents’ answers to 18 statements regarding noseband use. Statements 1–8 reveal the respondents ‘attitude’ towards noseband use, statements 9–13 reveal the ‘social factor (peer pressure)’, statements 14–18 reveal the ‘perceived behavioural control’, when applying the Theory of Planned Behaviour [10].

Statements Regarding Noseband Use		Cluster 1	Cluster 2	Cluster 3
*1*	It is more comfortable for the horse to be ridden without a noseband	Disagree	Disagree	Disagree
*2*	The use of a noseband is necessary to be able to ride a horse correctly	Neutral	Disagree	Disagree
*3*	I prefer the noseband fit tightly, so my horse does not open its mouth and the bit lies still, instead of the noseband applied too loose and my horse can open its mouth and the bit is moving around in the mouth	Agree	Disagree	Disagree
*4*	I’d rather my horse opens its mouth once in a while than applying the noseband too tight	Neutral	Agree	Agree
*5*	Riding with a tightly fitted noseband is better for the welfare of a horse	Agree	Disagree	Disagree
*6*	The position of the noseband has a greater impact on the horse than the extent to which the noseband is tightened	Neutral	Neutral	Neutral
*7*	The rider’s hand has a greater impact on the horse than the extent to which the noseband is tightened	Agree	Agree	Neutral
*8*	A looser noseband with a poorly fitting bit is just as uncomfortable for the horse as a tighter noseband with a well-fitting bit	Agree	Agree	Neutral
*9*	Riders deliberately tighten the noseband too tight	Neutral	Agree	Agree
*10*	I will approach someone who has tightened the noseband of the horse too tight	Disagree	Neutral	Neutral
*11*	I care if someone tells me the noseband of my horse is too tight	Neutral	Agree	Agree
*12*	I will tighten the noseband of my horse if my instructor tells me the noseband is too loose	Neutral	Neutral	Disagree
*13*	I do care if a judge tells me the noseband of my horse is too tight	Agree	Agree	Agree
*14*	The new regulations on noseband use leave room for discussion	Agree	Neutral	Neutral
*15*	Measuring should be done with a measuring instrument instead of with fingers	Agree	Neutral	Agree
*16*	The standard for the noseband should be adjusted to one finger instead of two fingers	Agree	Neutral	Disagree
*17*	The judge should be more moderate in judging a horse with an open mouth now that the new regulations have entered into force	Agree	Agree	Neutral
*18*	If the judge no longer deducts points for the horse riding with an open mouth or riding with the tongue out, riders would loosen the noseband	Agree	Agree	Neutral

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
