# Peer review of "Practice of Noseband Use and Intentions towards Behavioural Change in Dutch Equestrians"

_animals, 2019, doi:10.3390/ani9121131_

Round 1

Reviewer 1 Report

Document with edits sent separately.

Well done.

Author Response

Response to Reviewer 1

Thank you for your kind words and helpful comments.

We have replaced and corrected the English words in the Introduction as was suggested. Moreover, we have replaced nasal bone with frontal nasal plane in the whole paper.

We have replaced and corrected the English as suggested in the Materials and Method section except line 158 in which was suggested to add ‘between’ after ‘of’. We argue that this should stay ‘of’ since the results show no relation between either of the three variables and the variable age of the rider.

Line 127 – This passage is meant as a ‘warning’ for the reader that tightness level is inversely related to real tightness.

We have replaced and corrected the English as suggested in the Result section. We refrain from using capital letters at start of axis title words.

We have replaced and corrected the English as suggested in the Discussion and Conclusion. Thank you for the valuable comment to mention if noseband tightness was mentioned as the competitions. This was not the case and this has been added in the Discussion.

Reviewer 2 Report

This study is interesting and certainly the equine community needs this information in order to address potential welfare concerns in ‘show’ animals. The study was well thought out as the authors tried to link noseband tightness with the user survey data. It is unfortunate that the survey itself was ‘dressage’ biased, since the authors found tighter nosebands in the show jumping / eventing community. It would make a stronger paper if more survey data from show jumping/ eventing participants could be included. Is there anyway the authors can collect this data, add it to the results and re-submit? This reviewer would suggest adding figures of the different types of nosebands so that all readers (including international readers) would know what each noseband style looks like. As an example, the authors referred to a ‘Mexican noseband’, however in our country we would call it a ‘Figure 8’. It is confusing that so many of the results are discussed in the Methods section. Please reorganize the paper so that the appropriate information is discussed under the appropriate heading. Either the authors or the editor needs to make sure spacings are consistent throughout the paper.

Specific Comments:

Line 10  Replace the tightness of nosebands with noseband tightness

Line 27  Delete the information on ‘trends’ in the abstract.

Line 41  Provide some references to justify the statement of ‘increase in regulations aimed at…’

Line 44   Is ‘for horses in particular’ necessary?

Line 80-82  Please better clarify the ½ finger, 1 finger etc. by providing some estimations in something measurable (cm). It is done later in the paper, but would be better earlier.

Line 104   ‘in the Dutch situation’, needs more clarity

Lines 105-111    The wording here is confusing. Your objectives needed to be more clearly stated. Try to minimize the number of filler words

Line 117   Define all of the levels (i.e. letter designations, B etc.) so that all readers will understand

Line 136    Delete ‘that’

Line 146   generalized linear model (GLM), and then use GLM throughout the rest of the paper

Line 144    Explain your statistics here, and do not include the results in this section. Set your statistical significance here (ex. P < 0.05), as well as ‘trends’. Then do not discuss results that are ‘near significance. Either they are or they aren’t.

Line 187    Why is ‘teaming’ in italics?  Also please explain what ‘teaming’ is.

Line 212-213   Cited results in the methods is a bit confusing.

Line 233    Define ‘significance’ and ‘trends’ and then delete references to ‘near significance’

Line 359    This line is either confusing or redundant and needs clarification

Figure 1   Statistical significance should be noted. If there is none, please state that.

Figure 2    Statistical significance should be noted. If there is none, please state that

References (lines 485-526)

Authors need to proofread and correct format and missing information in the references. As an example, line 487 has (accessed on …..but no date etc..   Reference 4, cited a study from the Proceedings of International Conference Equitation Science needs a date and location.

Author Response

Response to Reviewer 2

Thank you for your helpful comments and suggestions.

We fully agree that it would have made it stronger paper if we had more data of show jumping respondents. Unfortunately, we have no opportunity at this moment to collect this data but will keep this in mind if a possibility occurs.

Instead of adding figures of the types of nosebands we referred to the paper of Uldahl and Clayton (2019). Moreover, in our results we have categorized the different types of nosebands based on their function which is described in the Results and have analyzed with these categories.

Since we would like the reader to focus on the final results we initially decided to have some of the results in the statistical analysis section, which was needed to explain how the final model was developed. We agree, that this might be confusing and therefore we moved these parts to the Results as was suggested.

We have checked for double spacings throughout the paper, thank you for mentioning.

Specific comments:

Line 11 – replaced

Line 27 – changed to  …was dependent on the interaction of competition level and discipline

Line 41 – we do not have a reference for this sentence, however we believe this is more or less common knowledge, so no reference needed.

Line 44 – we deleted in particular

Line 80 – thank you, we clarified that 2 fingers is approximately 1.5 cm

Line 104 – we changed into: Currently lack of info in the Netherlands

Line 105-111 sentences have been improved

Line 117 all letter have been defined at several places throughout the paper, thank you for the suggestion

Line 136 deleted, thanks

Line 146 Generalized Linear Model has been replaced by GLM throughout after the first

Line 144 This is clarified and moved to the results section

Line 187 – typo error, thank you – the sentence was redundant and has been deleted

Line 212-213 have been clarified and moved to results

Line 233 Significance is now defined at an earlier stage, we do not define trends

Line 359 sentence was redundant and has been deleted

Figure 1 – statistical significance is added in the figure caption

Figure 2 – statistical significance is added in the figure caption

References – we have corrected the references and added dates and places.

Round 2

Reviewer 2 Report

I appreciate the authors' response and quick changes to the article. I understand that it may be very difficult to increase the survey respondents to include more show jumpers and eventers. The authors recognize this as a concern, which cold be addressed in additional studies. Overall the manuscript is greatly improved regarding clarity. A few additional changes are recommended.
